# Experimental demonstration of adversarial examples in learning topological phases

Huili Zhang [1,4], Si Jiang [1,4], Xin Wang[1,4], Wengang Zhang[1], Xianzhi Huang[1,2], Xiaolong Ouyang[1], Yefei Yu[1], Yanqing Liu[1], Dong-Ling Deng[1,3] ✉ & L.-M. Duan [1] ✉

Classification and identification of different phases and the transitions between them is a central task in condensed matter physics. Machine learning, which has achieved dramatic success in a wide range of applications, holds the promise to bring unprecedented perspectives for this challenging task. However, despite the exciting progress made along this direction, the reliability of machine-learning approaches in experimental settings demands further investigation. Here, with the nitrogen-vacancy center platform, we report a proof-of-principle experimental demonstration of adversarial examples in learning topological phases. We show that the experimental noises are more likely to act as adversarial perturbations when a larger percentage of the input data are dropped or unavailable for the neural network-based classifiers. We experimentally implement adversarial examples which can deceive the phase classifier with a high confidence, while keeping the topological properties of the simulated Hopf insulators unchanged. Our results explicitly showcase the crucial vulnerability aspect of applying machine learning techniques in experiments to classify phases of matter, which can benefit future studies in this interdisciplinary field.

Machine learning, or more generally speaking artificial intelligence, is currently taking a technological revolution to modern society and becoming a powerful tool for fundamental research in multiple disciplines[1,2]. Recently, machine learning has been adopted to solve challenges in condensed matter physics[3–5], and in particular, to classify phases of matter and identify phase transitions[6–11]. Within this vein, both supervised[11–15] and unsupervised learning[7,16–21] methods have been applied, enabling identifying different phases directly from raw data of local observables, such as spin textures and local correlations[11–13]. In addition, pioneering experiments have also been carried out with different platforms[22–25], including electron spins in nitrogen-vacancy (NV) centers in diamond[22], cold atoms in optical lattices[23,24], and doped $CuO_2$[25], showing unparalleled potentials of machine learning approaches compared to traditional means.

An intriguing advantage of machine learning approaches in identifying phases of matter is that they may require only a small portion of data samples, without too much prior knowledge about the phases[22]. Therefore, they may substantially reduce the experiment cost in practice and are particularly suitable for exploring unknown exotic phases. However, the existence of adversarial examples[26–30], which can deceive the learning model at a high confidence level, poses a serious concern about the reliability of machine-learning approaches as well. The study of whether adversarial examples are potential obstacles in the experimental settings, and the experimental demonstration of adversarial examples in learning phases of matter, are still lacking hitherto. In this work, we find that the experimental noises are more likely to act as adversarial perturbations when a larger percentage of the input data are dropped or unavailable for the neural network-based classifiers. We present an experiment to implement adversarial examples and study their properties with a solid-state quantum simulator consisting of a single NV center in a diamond.

[1]Center for Quantum Information, IIIS, Tsinghua University, Beijing 100084, P. R. China. [2]School of JiaYang, Zhejiang Shuren University, Hangzhou 310015, P. R. China. [3]Shanghai Qi Zhi Institute, 41th Floor, AI Tower, No. 701 Yunjin Road, Xuhui District, Shanghai 200232, China. [4]These authors contributed equally: Huili Zhang, Si Jiang, Xin Wang. ✉e-mail: dldeng@tsinghua.edu.cn; lmduan@tsinghua.edu.cn

The NV center in diamond is a point defect[31], consisting of a nitrogen atom that substitutes a carbon atom and a nearest-neighbor lattice vacancy, as shown in Fig. 1a. It has long coherence time at room temperature and can be conveniently manipulated through lasers or microwaves, making this system versatile for applications in quantum networks[32–35], high-resolution sensing[36–39], quantum information processing[40–44], and quantum simulation[45–47], etc. Here, with the NV center platform, we experimentally demonstrate adversarial examples in learning topological phases, with a focus on the peculiar Hopf insulators[48,49]. More concretely, we first train a phase classifier based on deep convolutional neural networks (CNNs) so that it can correctly classify experimentally implemented legitimate samples with confidence close to one. We then show even though our legitimate samples have high fidelity (99.7% on average), when some data samples are dropped randomly, the tiny experimental noise can significantly affect the performance of the classifier. We also experimentally demonstrate that even without data dropping, the phase classifier based on neural network could be unreliable: after adding a tiny amount of carefully designed perturbations to the model Hamiltonian, the phase classifier would misclassify the experimentally generated adversarial examples with a confidence level up to 99.8%. The fidelity between the legitimate and corresponding adversarial samples is large (the average fidelity is 93.4%), ensuring that the adversarial perturbations added are small indeed. In addition, we extract the topological invariant and topological links by traditional means and demonstrate that they are robust to adversarial perturbations. This sharp robustness contrast between the traditional methods and machine-learning approaches clearly showcases the vulnerability aspect of the latter, highlighting the demand for in-depth investigations about the reliability of machine-learning approaches in adversarial scenarios and for developing countermeasures.

## Results

### Machine learning of topological phases

To experimentally implement adversarial examples and demonstrate the vulnerability of machine learning in topological phases, we first train a phase classifier based on deep neural networks to predict topological phases with high accuracy. Concretely, we focus on an intriguing three-dimensional (3D) topological insulator called the Hopf insulator[48,49], whose Hamiltonian in the momentum space reads:

$$H_{TI} = \sum_{\mathbf{k} \in BZ} \Psi_{\mathbf{k}}^{\dagger} H_{\mathbf{k}} \Psi_{\mathbf{k}} = \sum_{\mathbf{k}} \Psi_{\mathbf{k}}^{\dagger} \mathbf{u}_{\mathbf{k}} \cdot \boldsymbol{\sigma} \Psi_{\mathbf{k}}, \quad (1)$$

where $\mathbf{u}_{\mathbf{k}} = (u_x, u_y, u_z)$ with $u_x = 2(\sin k_x \sin k_z + C_{\mathbf{k}} \sin k_y)$, $u_y = 2(C_{\mathbf{k}} \sin k_x - \sin k_y \sin k_z)$, and $u_z = \sin^2 k_x + \sin^2 k_y - \sin^2 k_z - C_{\mathbf{k}}^2$; $\Psi_{\mathbf{k}}^{\dagger} = (a_{\mathbf{k},\uparrow}^{\dagger}, a_{\mathbf{k},\downarrow}^{\dagger})$ are fermionic annihilation operators with pesudo-spin states $|\uparrow\rangle$ and $|\downarrow\rangle$ at each momentum $\mathbf{k}$ in the Brillouin zone (BZ); $\boldsymbol{\sigma} = (\sigma_x, \sigma_y, \sigma_z)$ are Pauli matrices, $C_{\mathbf{k}} \equiv \cos k_x + \cos k_y + \cos k_z + h$. Hopf insulators are peculiar 3D topological insulators that originate from the mathematical theory of Hopf fibration and elude the standard periodic table for topological insulators and superconductors for free fermions[50,51]. They can manifest the deep connection between knot theory and topological phases of matter in a visualizable fashion[47,52]. Their topological properties can be characterized by a topological invariant (the Hopf index) $\chi$[48,49], defined as:

$$\chi = -\int_{BZ} \mathbf{F} \cdot \mathbf{A} d^3 \mathbf{k}, \quad (2)$$

where $\mathbf{F}$ is the Berry curvature defined as $F_\mu = \frac{1}{8\pi} \epsilon_{\mu\nu\tau} \hat{\mathbf{u}} \cdot (\partial_\nu \hat{\mathbf{u}} \times \partial_\tau \hat{\mathbf{u}})$ with $\hat{\mathbf{u}}(\mathbf{k}) \equiv \mathbf{u}(\mathbf{k})/|\mathbf{u}(\mathbf{k})|$, $\epsilon_{\mu\nu\tau}$ being the Levi-Civita symbol, and $\partial_{\nu,\tau} \equiv \partial_{k_{\nu,\tau}} (\nu, \mu, \tau \in \{x, y, z\})$; $\mathbf{A}$ denotes the Berry connection obtained by Fourier transforming $\nabla \times \mathbf{A} = \mathbf{F}$ with the Coulomb gauge $\nabla \cdot \mathbf{A} = 0$

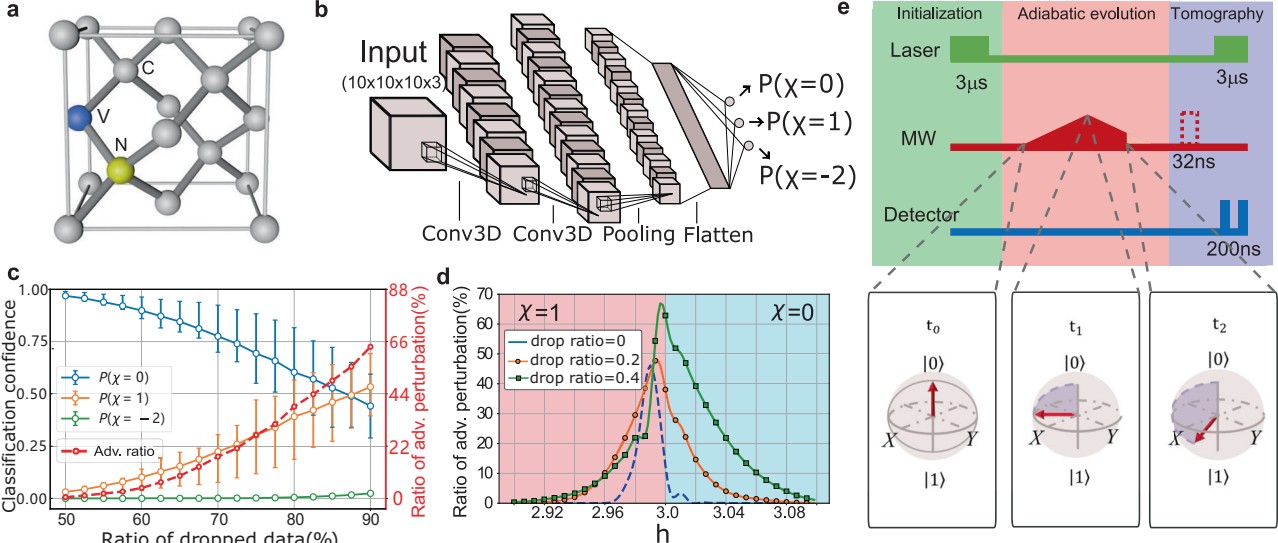

**Fig. 1 | Experimental setup and the topological phase classifier. a** The structure of a nitrogen-vacancy center in a diamond crystal. The blue, yellow, and gray spheres represents vacancy, nitrogen, and carbon atoms, respectively. **b** The structure of 3D convolutional neural network (CNN) classifier. The input data are density matrices sampled from a $10 \times 10 \times 10$ regular grid in the momentum space, and each density matrix is represented by three real indices among the Bloch sphere. With two 3D convolution (Conv3D) layers, one max pooling layer and one fully connected layer, the classifier outputs the probabilities $P(\chi = 0, 1, -2)$ for each phase. **c** The potential limitation of the CNN classifier on data with random dropping. The classifier can correctly classify the clean data of topologically trivial phase ($h = 3.2, \chi = 0$) with more than 80% of the data samples dropped, but the adversarial

ratio (Adv. ratio) also increases as the dropping ratio increases. The error bars are obtained from 100 random data dropping trials. **d** The ratio of adversarial perturbations around the phase transition point. The random simulated perturbations are more likely to behave as adversarial perturbations when $h$ approaches the transition point. Even when no data samples are dropped, the simulated perturbations may mislead the classifier. The situation becomes more serious when the dropping ratio increases to 20% and 40%. **e** The experimental procedure for the preparation and measurement of the ground states of the Hopf Hamiltonian at each momentum $\mathbf{k}$. The dashed rectangle inserted before the final measurement represents a $\pi/2$ pulse with different phases. The directions of the electron spin on the Bloch sphere at three different time points are shown below the sequence.

(see Methods). For the Hamiltonian $H_{TI}$, direct calculations yield $\chi = -2$ if $|h| < 1$, $\chi = 1$ if $1 < |h| < 3$, and $\chi = 0$ otherwise.

We numerically generate 5000 samples with varying $h \in [-5, 5]$ and train a 3D CNN in a supervised fashion. Figure 1b shows the structure of the CNN classifier we use. The 3D CNN model, which has three-dimensional structure and presents the translational symmetry, is suitable for learning phases with Bloch indices of density matrices in the momentum space. The input data are reconstructed density matrices in the momentum space. The classifier's outputs are classification confidences for the phase being $\chi = 0$, 1, or $-2$, denoted by $P(\chi = 0)$, $P(\chi = 1)$, or $P(\chi = -2)$, respectively. After training, the classifier obtains near-perfect performance on the numerically generated data, with accuracies of 99.2% and 99.6% on the validation and training sets, respectively (see Supplementary Note 5).

As shown in ref. 22, a well trained CNN classifier can correctly identify the phases with high ($\geq 90\%$) probability even when over 90% of data are randomly dropped. This would significantly reduce the experimental cost since much fewer data samples are required. However, the existence of experimental noises is a challenge for the CNN models to showcase this advantage in reality. To test the robustness of the classifier on dropped data with noise, we numerically simulate 1000 tiny perturbations which are randomly sampled from the Gaussian distribution. We find that some of the simulated perturbations, when added into the legitimate samples, will mislead the classifier. Such perturbations, though not carefully crafted, behave like adversarial perturbations as well. As shown in Fig. 1c, although the classifier can correctly identify the phases on the clean data with more than 80% of density matrices in the discretized momentum space dropped, the ratio of adversarial perturbations also increases rapidly. This problem becomes more serious when the testing samples are closer to the phase transition point: as shown in Fig. 1d, dropping only 40% of the data will increase the ratio of adversarial perturbations to about 68%, which makes the precise identification of phase transition points unpractical.

## Experimental implementation

We use a single NV center as a simulator to experimentally implement the model Hamiltonian in Eq. (1). The ground state of the NV electron spin consists of $|m_s = 0\rangle$ and degenerate $|m_s = \pm 1\rangle$ state with zero-field splitting of 2.87 GHz[53,54]. Our setup is based on a home-built confocal microscope with an oil-immersed lens. To enhance photon collection efficiency, a solid immersion lens is fabricated on top of the NV center. A magnetic field of 472 Gauss is applied along the NV symmetry axis to polarize the nearby nuclear spins and remove degeneracy between states $|m_s = \pm 1\rangle$. We use the subspace $|m_s = 0\rangle$ and $|m_s = -1\rangle$, denoted as $|0\rangle$ and $|-1\rangle$ of the electron spin in the experiment. The experiment sequence is shown in Fig. 1e. The spin state is initialized to $|0\rangle$ by optical pumping. Then, a microwave (MW) is applied to adiabatically evolve the spin state. By tuning the amplitude, frequency, and phase of MW, the electron spin is evolved to the ground state of the corresponding Hamiltonian at a given momentum point $\mathbf{k}$[22,55]. The electron spin states at three different evolution time points are shown below the sequence in Fig. 1e. After the adiabatic evolution, quantum state tomography is performed, and the state density matrices are retrieved via maximum likelihood estimation[56].

To obtain density matrices sampling of the Hamiltonian in Eq.(1), we mesh the momentum space $\mathbf{k} = (k_x, k_y, k_z)$ into $10 \times 10 \times 10$ grids with equal spacing. We use ground state density matrices of $H_{\mathbf{k}}$ with $h = 0.5, 2, 3.2$ as legitimate samples. These legitimate samples are used as ground truth to evaluate the topological phase classifier and latter generate corresponding adversarial examples. To alleviate the effect of experimental Gaussian noise on misleading the classifier, we implement all reconstructed states with a very high fidelity: for all three legitimate samples with $h = 3.2, 2, 0.5$, the average fidelities are 99.77(41)%, 99.78(41)%, and 99.77(45)%, respectively (for the fidelity on

## Table 1 | The classifier's prediction results

| Input | | $P(\chi = 0)$ | $P(\chi = 1)$ | $P(\chi = -2)$ |
|---|---|---|---|---|
| $h = 0.5$ | Leg. | 0 | 0 | 1 |
| | Adv.(C.) | 0 | 0.776 | 0.224 |
| $h = 2.0$ | Leg. | 0 | 1 | 0 |
| | Adv.(C.) | 0.004 | 0.293 | 0.703 |
| $h = 3.2$ | Leg. | 1 | 0 | 0 |
| | Adv.(C.) | 0.002 | 0.998 | 0 |
| | Adv.(D.) | 0.262 | 0.738 | 0 |

The output on experimentally implemented legitimate (Leg.) samples and their corresponding adversarial (Adv.) examples. The classifier successfully identifies the phase label for legitimate samples with nearly unity confidence. Yet, it is deceived by experimentally implemented adversarial examples, making incorrect predictions with confidence larger than 0.5. C. (D.) represents adversarial examples obtained by continuous(discrete) attacks.

each momentum point, see Supplementary Fig. 2). The table in Table 1 presents the classifier's output for three legitimate samples, which are all correct classifications with nearly unity confidence.

Although the experimental noises in the implemented legitimates samples are tiny, we find that they can still affect the classifier's performance when we try to use machine learning methods to reduce experimental data. As shown in the lower panel of Fig. 2a, we randomly drop the experimentally implemented density matrices of the legitimate sample with $h = 3.2$. We find that the experimental noises will decrease the classification confidence $P(\chi = 0)$ as the ratio of dropped data increases, and act as adversarial perturbations when the dropping ratio increases to 60%. This result is consistent with the numerical simulation, indicating that there is a trade-off for the machine learning methods between the number of data samples used and the robustness to experimental noises. This trade-off cannot be attributed to the decreasing of the quality of the experimental data: we show the fidelity distribution in the upper panel of Fig. 2a, where the average fidelity $\bar{F}$ and minimum fidelity $F_{min}$ are almost unchanged when the dropping ratio increases. The phenomenon that the classifier would be misled by tiny perturbations is not limited to the case where a large portion of data is dropped. Actually, with the full data, there are various kinds of tiny perturbations that can make the classifier give incorrect predictions, which is known as the vulnerability of neural networks to adversarial perturbations. This may rise severe problems for machine-learning approaches to the classification of different phases of matter, which drives us to experimentally implement these adversarial examples and study their topological properties.

We now consider implementing adversarial examples by adding tiny adversarial perturbations without data dropping. Choosing the loss function $L$ as the metric to evaluate the performance of the classifier, we first search for numerical adversarial examples which can mislead the classifier by solving an optimization problem[30]: finding a bounded perturbation $\delta$ adding to the legitimate samples' data to maximize the loss function $L$. We employ various strategies to approximately solve this optimization problem, including the fast gradient sign method (FGSM)[57], projected gradient descent (PGD)[57], momentum iterative method (MIM)[58] and differential evolution algorithm (DEA)[59–61]. More concretely, we apply PGD and MIM to obtain continuous adversarial perturbations based on all three legitimate samples[62], and apply DEA to obtain one adversarial example with discrete perturbations based on the legitimate sample with $h = 3.2$ (see Supplementary Note 6).

We remark that the existence of numerical adversarial examples does not guarantee that we can implement them in real experiments due to inevitable experimental imperfections—the experimental noises may wash out the bounded and carefully crafted adversarial perturbations. In fact, as shown in refs. 63,64, certain noises that are typical in experiments would nullify the adversarial examples. In the worst case, when the noise is in the opposite direction of the adversarial

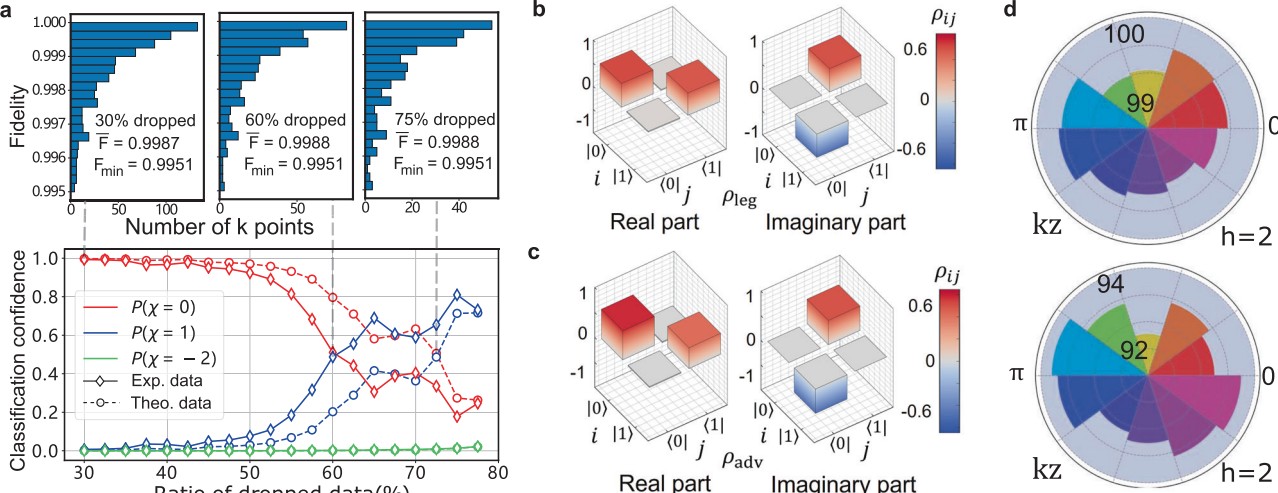

**Fig. 2 | Experimentally implemented adversarial examples. a** The classification confidence for experimental (Exp.) and theoretical (Theo.) data at $h = 3.2$ with different dropping ratios. For theoretical (experimental) data, the classifier gives correct predictions with 75% (60%) of the data samples dropped. The panels in the upper row show the fidelity distribution of the remaining data when the dropping ratios are 30%, 60%, and 75%, respectively. The average fidelity $\bar{F}$ and minimum fidelity $F_{\min}$ remain almost unchanged at different dropping ratios. **b** Density matrix of the experimentally implemented legitimate sample $\rho_{\text{leg}}$ at $\mathbf{k} = (0.2\pi, 0.6\pi, 1.4\pi)$ and $h = 0.5$, with fidelity

99.68(31)%. **c** Density matrix of the experimentally implemented adversarial example $\rho_{\text{adv}}$ at the same parameter point as in **b**, with fidelity 99.23(26)%. **d** Upper panel: the average fidelity for each $k_z$ value in the interval $[0, 1.8\pi]$ with $h = 2$ between numerically generated and experimentally implemented adversarial examples. The angular direction represents the different values of $k_z$ and the radial direction represents the fidelity. The overall average fidelity is 99.65(46)%; Lower panel: the average fidelity between experimentally implemented legitimate samples and adversarial examples. The overall average fidelity is 93.40%.

perturbation, the experimentally implemented adversarial examples will no longer be able to mislead the classifier. To this end, we numerically simulate the experimental noises acting on these numerically obtained adversarial examples and examine their performances on the classifier. After the simulation, for each scenario we select one example with the strongest robustness against experimental noises and reconstruct its corresponding Hamiltonian. The comparison between densities for the experimentally implemented legitimate and adversarial examples are shown in Fig. 2b, c. Figure 2d shows the average fidelity for each $k_z$ value with $h = 2$. All adversarial examples bear high fidelities (larger than 93%), but the classifier incorrectly predicts their phase labels with a high confidence level, as shown in Table 1.

### Demonstration of adversarial examples

In the previous section, we illustrate that the experimentally implemented adversarial examples, which maintain a high fidelity with respect to original legitimate data, can mislead the topological phase classifier. In this section, we further demonstrate the effectiveness of these adversarial examples from the physical perspective. In Fig. 3a–c, with $h = 3.2$, we plot experimentally implemented density matrices of legitimate samples, adversarial examples with continuous perturbations, and adversarial examples with discrete perturbations. From the comparison, the obtained adversarial examples look almost the same as the original legitimate ones. It is surprising that even local discrete changes, as shown in Fig. 3c, can mislead the classifier to make incorrect prediction. This result is at variance with the physical intuition that Hopf insulators are robust to local perturbations due to their topological nature[48,49], indicating that the neural network based classifier does not fully captured the underlying topological characteristics[30].

Focusing on classifying topological phases of Hopf insulators, we expect that the adversarial perturbations should not change the topological properties, including the integer-valued topological invariant and the topological links associated with the Hopf fibration[48]. We use a conventional method, which is based on the experimentally measured data, to probe the Hopf index[65,66]. The results are shown in

Table 2 (see Supplementary Note 4). We find that each adversarial example's Hopf index has only a negligible difference to the corresponding legitimate ones, all close to the correct integer numbers.

The momentum-space spin texture of Hopf insulators harbors a knotted structure, which is called the Hopfion[67]. In Fig. 3d–f, we show cross sections of the measured spin textures before and after adding continuous and discrete adversarial perturbations. The spin textures present an illustration on 3D twisting of the Hopfion, which keep original structure and are almost not affected by adversarial perturbations. A more intuitive demonstration of Hopf links can be derived if we consider the preimage of a fixed spin orientation on the Bloch sphere, which will form a closed loop in the momentum space $\mathbb{T}^3$. For topological nontrivial phases, the loops for different orientations are always linked[47]. As shown in Fig. 4, we plot the 3D preimage contours of legitimate and adversarial examples in $\mathbb{R}^3$ in the stereographic coordinates of $\mathbb{S}^3$, with $h = 0.5$, orientations $\mathbf{S} = (-1, -1, 0)/\sqrt{2}$ and $(0, 1, -1)/\sqrt{2}$ on the Bloch sphere (see Supplementary Note 7). We observe that the loops are correctly linked together as $h = 0.5$ corresponds to topological nontrivial $\chi = -2$ phase, for both legitimate and adversarial examples. This result illustrates that the adversarial perturbations do not affect the Hopf link, despite the fact that they alter the predictions of the classifiers drastically.

A possible defense strategy against adversarial perturbations is adversarial training. The basic idea is to retrain the classifier with a new training set that contains both legitimate and adversarial samples[28]. Here, we retrain the phase classifier with both carefully crafted adversarial examples and samples with experimental noises. We find that this adversarial training strategy can indeed substantially enhance the robustness of the classifier against adversarial perturbations and experimental noises. Yet, after adversarial training the classifier's performance near the phase transition points becomes poorer. This is attributed to the fact that adversarial training will flatten weight parameters in general (See Supplementary Note 8).

## Discussion

The above sections showcase that the adversarial perturbations do not affect the topological properties of topological phases. The incorrect

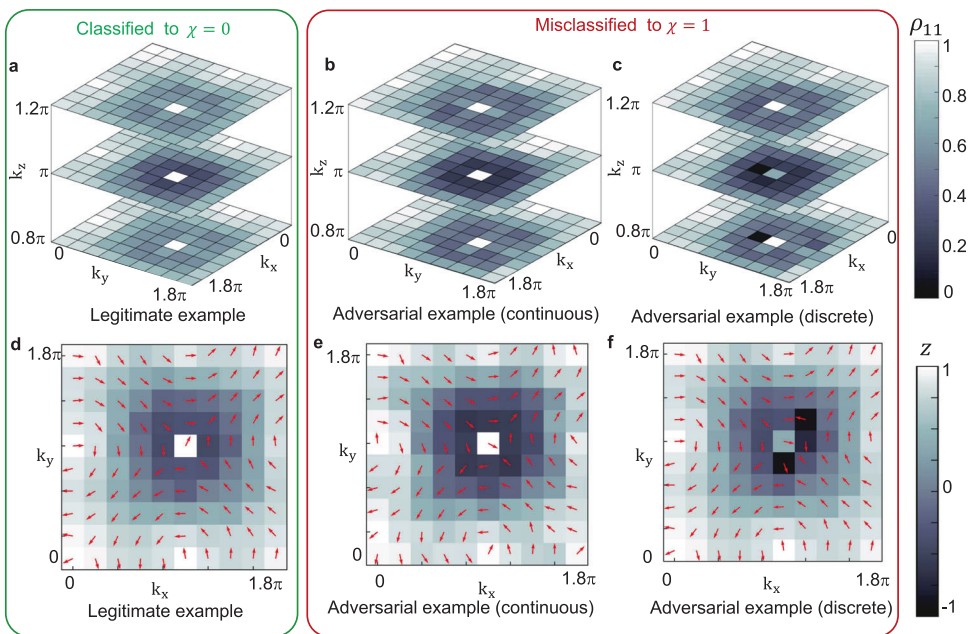

**Fig. 3 | Visualization of experimentally realized density matrices for the Hopf insulators with h = 3.2. a–c** The first component's magnitude of the input data for $h = 3.2$ at $k_z = 0.8\pi$, $\pi$ and $1.2\pi$. **a** Legitimate sample with $h = 3.2$ implemented in the experiment. **b** Adversarial examples realized in the experiment with continuous perturbations generated by the projected gradient descent method. The average fidelity between the experimentally implemented adversarial examples and legitimate samples is over 98%. **c** Adversarial examples realized in the experiment with discrete perturbation generated by the differential evolution algorithm. Among 1000 density matrices as input, only seven of them have been changed and successfully mislead the classifier. **d–f** Measured spin texture for $k_z = \pi$, $h = 3.2$. For each subfigure, $k_x$ and $k_y$ vary from 0 to $1.8\pi$ with equal spacing of $0.2\pi$. At each momentum **k**, the state can be represented on the Bloch sphere. The arrows in the plane show the direction of the Bloch vector projected to the $x - y$ plane. The colors label the $z$ component of the Bloch vector. **d** Legitimate sample with $h = 3.2$ implemented in experiment. **e** Adversarial examples implemented in the experiment with continuous perturbations generated by the momentum iterative method. **f** Adversarial examples implemented in the experiment with discrete perturbation generated by the differential evolution algorithm.

predictions given by the classifier indicate that the classifier does not learn the accurate and robust physical criterion for identifying topological phases, which is consistent with the theoretical prediction in the recent paper[30]. How to exploit the experimental data to make the classifier better learn physical principles, and balance the trade-off between the number of data samples used and the classification accuracy in real experiments, is an interesting and important problem worth further investigation. In addition, recent experiments have demonstrated the simulation of non-Hermitian topological phases with the NV platform[68] and a theoretical work on learning non-Hermitian phases with exotic skin effect in an unsupervised fashion has also been reported[17]. In the future, it would be interesting and desirable to study adversarial examples for unsupervised learning of topological phases, both in theory and in experiment.

In summary, we have experimentally demonstrated the adversarial examples in learning topological phases with a solid-state simulator. Our

result showcases that neural network-based classifiers are vulnerable to tiny adversarial perturbations, which may originate from experimental noises, or be carefully designed. In the former case, we showed that there is a trade-off between the number of data sample used and the robustness to tiny experimental noises. For the latter case, we implemented the adversarial examples in experiment and studied their properties, such as the high fidelity, unchanged topological invariant, and topological link. These results reveal that current machine learning methods do not fully capture the underlying physical principles and thus are especially vulnerable to adversarial perturbations, and inevitable experimental noises when only a small portion of data are accessible.

## Methods

### Experimental setup

Our experiment is implemented on a home-built confocal microscope at room temperature. The 532 nm diode laser passes through an acoustic optical modulator setting in a double-pass configuration. The laser can be switched on and off on the time scale of ~20 ns with on-off ratio to 10,000:1. A permanent magnetic provides the static magnetic field of 472 Gauss. The magnetic field is precisely aligned parallel to the symmetry axis of the NV center by observing the emitted photon numbers[69]. With this magnetic field, a level anticrossing in the electronic state allows electron-nuclear spin flip-flops, which polarizes the nuclear spin[70]. The magnetic field also removes the degeneracy between $|m_s = \pm 1\rangle$ states. The spin state is initialized by a $3\mu s$ laser excitation, then a MW modulation is implemented by programming two orthogonal 100 MHz carrier signals, which are generated by arbitrary waveform generator. The MW is amplified and guided through coaxial cables to gold coplanar waveguide close to the NV center. The emitted photons are collected through an oil-immersed objective lens (NA = 1.49) and detected by an avalance photodiode.

**Table 2 | The Hopf index calculated by using the conventional discretization approach**

| Input | h = 0.5 | h = 2 | h = 3.2 |
|---|---|---|---|
| Theory ($N \to \infty$) | −2 | 1 | 0 |
| Theory ($N = 10$) | −2.045 | 1.041 | 0.011 |
| Leg. samples | −2.056 | 1.039 | 0.009 |
| Adv. examples (C.) | −2.039 | 0.952 | 0.026 |
| Adv. examples (D.) | – | – | 0.073 |

For a $10 \times 10 \times 10$ grid, the theoretical value for the Hopf index is shown in the third row, where the small deviations from the corresponding integers are caused by the discretization error in the 3D momentum integration. The Hopf indices of experimentally implemented legitimate (Leg.) samples, and adversarial (Adv.) examples with continuous (C.) and discrete (D.) attacks are close to the theoretical values.

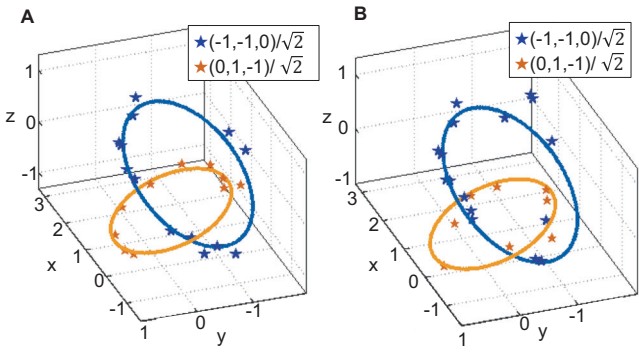

**Fig. 4 | The 3D preimage contours which show topological links for the Hopf insulator with** h = 0.5. **a** Topological link, obtained from two spin states with Bloch sphere representations $\mathbf{S} = (-1, -1,0)/\sqrt{2}$ (blue) and $(0,1, -1)/\sqrt{2}$ (orange), for a legitimate sample in the stereographic coordinates of $\mathbb{S}^3$, between two spin states on the Bloch sphere. Solid lines are curves from numerically calculated directions $\mathbf{S}_{th}$ and the stars are experimentally measured spin orientations $\mathbf{S}_{exp}$. The deviation $|\mathbf{S}_{exp} - \mathbf{S}_{th}| \leq 0.25(0.35)$ for the blue (orange) curve. **b** The topological link for an adversarial example with deviation $|\mathbf{S}_{exp} - \mathbf{S}_{th}| \leq 0.25$ for blue and orange curves.

The spin state is readout by counting the spin-dependent number of photons. To enhance collection efficiency, a solid immersion lens with 6.74 μm diameter is fabricated. The fluorescence count is about 260 kcps under 0.25 mW laser excitation, with the signal-noise ratio about 100:1. The sequence is repeated $7.5 \times 10^5$ times, collecting about $3.9 \times 10^4$ photons.

### Adiabatic passage approach

Consider the electron subspace spanned by the $|0\rangle$ and $|-1\rangle$ states, in a rotating frame, the effective Hamiltonian with variable time $t$ reads

$$H_{\text{eff}} = \begin{pmatrix} 0 & |\Omega(t)|e^{i\varphi} \\ |\Omega(t)|e^{-i\varphi} & -\Delta\omega(t) \end{pmatrix}, \tag{3}$$

where $\Omega(t)$ is the MW amplitude, $\varphi$ is the MW phase, $\Delta\omega(t) = \omega_0 - \omega_{\text{MW}}$, and $\omega_0$ and $\omega_{\text{MW}}$ are NV resonant frequency and MW frequency, respectively. In the matrix form, the Hamiltonian $H_{\mathbf{k}}$ reads:

$$H_{\mathbf{k}} = \begin{pmatrix} 0 & u_x - iu_y \\ u_x + iu_y & -u_z \end{pmatrix}. \tag{4}$$

We terminate the adiabatic evolution at time $t_c$ to satisfy $\Delta\omega(t_c)/\Omega(t_c) = u_z/\sqrt{u_x^2 + u_y^2}$. Phase $\varphi = -\arctan(u_y/u_x)$ is kept constant in the adiabatic evolution. To satisfy the adiabatic condition[22]:

$$|\frac{\hbar\langle \psi^{(e)}|\dot{\psi}^{(g)}\rangle}{E_e - E_g}| \ll 1, \tag{5}$$

where $|\psi^g\rangle$ and $|\psi^e\rangle$ are ground and excited states of the Hamiltonian $H_{\text{TI}}$, and $E_g$ and $E_e$ are energies of the ground and excited states, respectively. We use $\Omega_{\max} = 2\pi \times 7.81$ MHz and $\Delta\omega_{\max} = 2\pi \times 10$ MHz during the adiabatic passage process for a total time of 1500 ns.

### Conventional method to obtain the Hopf index with experimental data

We use the discretization scheme introduced in refs. 65,66 and applied in ref. 47 to calculate the Hopf index directly from experimental data. The Hopf index can be written as:

$$\chi = -\int_{\text{BZ}} \mathbf{F} \cdot \mathbf{A} d^3\mathbf{k}, \tag{6}$$

where $\mathbf{F}$ is the Berry curvature with and $\mathbf{A}$ is the associated Berry connection satisfying $\nabla \times \mathbf{A} = \mathbf{F}$. To avoid the arbitrary phase problem, we can use a discretized version of the Berry curvature[65,66]:

$$F_\mu(\mathbf{k}_J) = \frac{i}{2\pi}\epsilon_{\mu\nu\tau}\ln U_\nu(\mathbf{k}_J)\ln U_\tau(\mathbf{k}_{J+\hat{\nu}}). \tag{7}$$

The $U(1)$-link is defined as

$$U_\nu(\mathbf{k}_J) = \frac{\langle \psi(\mathbf{k}_J)|\psi(\mathbf{k}_{J+\hat{\nu}})\rangle}{|\langle \psi(\mathbf{k}_J)|\psi(\mathbf{k}_{J+\hat{\nu}})\rangle|}, \tag{8}$$

with $\hat{\mathbf{v}} \in \{\hat{\mathbf{x}},\hat{\mathbf{y}},\hat{\mathbf{z}}\}$, which is a unit vector in the corresponding direction. This discretized version of $\mathbf{F}$ can be calculated after performing quantum state tomography at all points $\mathbf{k}_J$ on the momentum grid. We can also obtain the Berry connection $\mathbf{A}$ by Fourier transforming $\nabla \times \mathbf{A} = \mathbf{F}$ with the Coulomb gauge $\nabla \cdot \mathbf{A} = 0$. Finally, instead of doing the integral, we sum over all points on the momentum grid to obtain the Hopf index $\chi$. It is shown in[22,47] that for a $10 \times 10 \times 10$ grid, this method is quite robust to various perturbations and can extract the Hopf index with high accuracy.

### The simulation of experimental noise and random dropping trials

To test the robustness of the neural network-based classifier against experimental noises and random data dropping, we numerically simulated 1000 tiny experimental noises and 100 different data point dropping sequences on the theoretically calculated data with $h = 3.2$ and $\chi = 0$. Each experimental noise is simulated by the $10 \times 10 \times 10 \times 3$ independent variables sampled from the normal distribution:

$$\epsilon \sim \mathcal{N}(0,0.04). \tag{9}$$

The noise is directly added on the Bloch vector for each density matrix. After this, we normalize the obtained vector so that it still represent a pure state on the Bloch sphere. The noise is tiny enough to keep the high fidelity ($\overline{F} = 0.9899$). The data dropping is implemented by sequentially replacing the Bloch vectors $(n_x^{(\mathbf{k})},n_y^{(\mathbf{k})},n_z^{(\mathbf{k})})$ at randomly selected $\mathbf{k}$ point by a zero vector $(0, 0, 0)$.

### Data availability

The data generated in this study have been deposited in the zenodo database under accession code https://doi.org/10.5281/zenodo.6830983.

### Code availability

The data analysis and numerical simulation codes are available at https://doi.org/10.5281/zenodo.6811855.

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

## Acknowledgements
This work was supported by the Frontier Science Center for Quantum Information of the Ministry of Education of China, Tsinghua University Initiative Scientific Research Program, and the Beijing Academy of Quantum Information Sciences. D.-L.D. also acknowledges additional support from the Shanghai Qi Zhi Institute.

## Author contributions
H.Z. and X.W. carried out the experiment under the supervision of L.-M.D. and D.-L.D. S.J. did the numerical simulations and analyzed the experimental data together with H.Z. W.Z., X.O., X.H., Y.Y., and Y.L. contributed to the fabrication of the diamond sample, and helped with the experimental measurements. All authors contributed to the experimental set-up, the discussions of the results and the writing of the manuscript.

## Competing interests
The authors declare no competing interests.

## Additional information
**Supplementary information** The online version contains

supplementary material available at https://doi.org/10.1038/s41467-022-32611-7.

