## [Peer Review File · Nature Communications]

REVIEWER COMMENTS

Reviewer #1 (Remarks to the Author):

Review of Experimental demonstration of adversarial examples in learning topological phases

In the paper by Zhang et al., Experimental demonstration of adversarial examples in learning topological phases, the authors show that state of the art CNN models, capable of classifying experimental data from an NV centre in diamond, are strongly affected by adversarial examples.

Adversarial examples are carefully crafted forms of noise obtained from the fully trained model that are superimposed to the experimental data. They trick the model into failing the predictions, as reported several times in literature, but require full access to the trained model and need to counterfeit the experimental data.

The topic is timely and of clear relevance for the ML community and also for people working in the characterization of quantum systems and quantum technologies. I enjoyed reading the article, and the main ideas are mostly, well presented, and scientific experimental results are of very high quality, in line with previous results from the same authors. In the absence of adversarial examples, their machine learning models show state of the art performance. Although, the paper could and should be improved to highlight their main claims.

From the scientific point of view, I have these major doubts:

1. If they claim that ML cannot be trusted in every scenario, I agree with the authors, but this would not be a novel result. The central claim and its importance need to be highlighted in order to assess properly the impact of their work.
2. They show vulnerability against adversarial examples, a very well-known problem, for a specific case in physics. But this requires access to the model, and if we train again on different data points, the same adversarial examples might not work. So it is not clear how this would impact the work in the lab of scientists characterizing quantum systems without being attacked and having their experimental data tampered with.
3. Is the use of tomography, a highly unscalable method, for the characterization of a solid-state system something the authors would expect to be applicable in a broader scenario? How would the

authors envisage the use of a similar methodology for larger systems? The number of data points would scale exponentially fast. This is a key point because if the technique is not scalable is never going to be applied in practical cases.

While I like the article and enjoyed reading it, I am not yet convinced that the results, as they are now, deserve publication in a high-impact journal such as Nature Communications. And I would be happy to be convinced that instead, they deserve publication in NC.

To properly assess the impact of this work, we need to ask the question, are adversarial examples a real threat for solid-state system characterization? It is unclear to me, in the current version of the manuscript, whether this is going to be the case or not.

Adversarial examples are a clear threat for example, for self-driving cars, where one could place a carefully crafted image printed on a board and placed on the street to make the car crash.

But in this specific study, what kind of scenario the authors are envisaging?

Do we expect somebody to steal the model used in the characterization process of the solid-state system and carefully falsify the experimental data to make it look like a system is in a phase rather than in another one?

The authors should clarify in what circumstances or scenarios the adversarial examples would be used. A random noise will not have the same effects as adversarial examples, and the literature in ML is full of examples where very small perturbations, if carefully crafted, can heavily affect the performance of ML algorithms.

In general, I agree with the authors we cannot trust CNN models to predict effectively against adversarial examples. This is a topic of research and an important one. Of course, if the authors provided a solution to the problem then it would be a different story, but in this case, I would be happy to have a convincing practical case where this problem would be of impact on scientific research.

Reviewer #2 (Remarks to the Author):

In "Experimental demonstration of adversarial examples in learning topological phases" the authors experimentally demonstrate the failure of a classification based on neural nets upon adversarial attacks. They use an NV center for the study of the dispersion of a 3D Hopf insulator. Based on the reconstructed density matrices they first train a CNN to predict the Hopf index, with great success. Then they perform an optimization problem to devise local perturbations that maximize the loss function and hence fool the neural net. They can implement these adversarial attacks in the experiment, and they find that the phase classification has become useless indeed. It follows that the neural net has not learned the correct physical property based on topology. It is unclear what was learned instead.

From the theoretical perspective, the conclusions in this work are not at all surprising. It is well known that a naive classifier fails to adequately capture topological information. But the field has moved on; there are now several papers addressing topological phases with machine learning, not only the ref 15-17 in the paper.

We also know what to do. Essentially, we need to know about the behavior of the curvature around high symmetry points such as $(0,0,0)$, $(0,0,\pi)$ etc. (so you should not need all k-points, you can reduce the input dimension!). It is possible to correctly classify certain topological phases based on this curvature.

The curvature function typically involves derivatives of the phase. The reduced density matrices, which serve as the input here, contain very little information about the phase. So one can equally say the problem lies with the humanly chosen input than with the neural network.

Experimentally the results are convincing. The reconstruction of the density matrices based on quantum state tomography and maximum likelihood estimation works well for this system, and are of course a convenient input for the neural net. The reported fidelities are very high and convincing, and the implementation of the adversarial attack is done with very high precision too. The text is very well written and can be understood by non-experts.

Although I am in general all in favor of publishing a negative result, I must say that this negative result here is totally expected. The sentence "How to exploit the experimental data to make the classifier better learn physical principles, and how to protect the neural-network based classifiers in real experiments from adversarial perturbations, are interesting and important problems worth further investigation" in the conclusion is however below my expectations: I would have expected more on this, given that the field has moved on and that ideas of classifying topology do exist in the literature.

Reviewer #3 (Remarks to the Author):

In their paper 'experimental demonstration of adversarial examples in learning topological phases', the authors study the Hopf model using a convolutional neural network. They employ both numerical simulation as well as experiments on an NV center to generate the quantum state of the system for different momenta and values of the Hamiltonian parameter h . They use different methods to find adversarial examples which will most confuse the neural network, and implement those examples both numerically as experimentally, and show that the neural network can indeed be deceived by those examples.

The general topic of adversarial examples is important when using machine learning tools for scientific insights, and I believe the authors made some progress in the present work. However, several things in the manuscript are unclear to me and therefore hard to judge.

- what is a Hopf insulator -- despite the paragraph on p.2 about this model, it is not clear to me why this model has topological phases of matter, or how this is related to the NV center described earlier, or how one arrives at the Hopf index. Since this seems to be (?) what the network is trained to predict, it should be explained in more detail, and its relation to the density matrix should be clarified.

- what is predicted by the neural network? This is a crucial question that should be very clear, but is not to me from the present text.

- Fig. 2h: it would be very useful if the reader knew how the Hopf index is obtained from the density matrix. In particular, how the density matrices for different momenta and different values of h are connected. Is it possible that a reordering in momentum space (possibly obtained through convolution and max pooling in the neural network?) leads to the wrong classification?

As the authors themselves write "This result is at variance with the physical intuition that Hopf insulators are robust to local perturbations due to their topological nature [45, 46], indicating that the neural network based classifier does not fully capture the underlying topological characteristics [27]." -- this makes it even more important to explain to the reader why the model should be robust to local perturbations, and what it actually is that they ask their network to predict. Without this information the results are very hard to appreciate. Given the next paragraph in the text, it is even less clear to me what they were classifying before.

- I don't really understand why it's important to experimentally engineer one particular adversarial example. Isn't it rather the other way round, that we should make sure that experimental imperfections don't lead to (undesired!) adversarial examples? This leads to the question of how fine-tuned these adversarial examples are. I'm assuming that this single qubit system is experimentally very well controlled, also given the fidelities in preparing a given quantum state demonstrated in the paper. In particular, given the fidelity of 99.23% between the numerical and experimental adversarial example, it doesn't seem that there are too many experimental errors in play here.

It's unclear what happens with a slightly less fine-tuned adversarial example -- does it still get misclassified? Or do we only get this strong effect if we choose one very very specific state? In my opinion, this makes a big difference for the importance of this problem.

- the importance and difference between continuous and discrete perturbations is not clear to me.

- Fig. 3: to really appreciate what's happening here and how surprising it is, it would be tremendously helpful to also show the corresponding images for an example from the phase (?? Or value of χ ?) AS WHICH the neural network wrongly classifies the adversarial example. This would greatly help to understand how close this adversarial example is to the 'right' vs 'wrong' phase. Another way to showcase this would be to calculate fidelities to all possible states from the 'wrong' phase, because in the end the question is whether these fidelities are low compared to the confidence of the network in its wrong classification. This very crucial piece of information is not clear to me from the paper.

Minor comments:

- Fig. 1a is a bit confusing -- unclear what's supposed to be 3D and what not -- where is the vacancy?

- Fig. 1 caption: "indexes" (indices?)

- Fig. 1c: it is not clear to me why a convolutional (as opposed to a fully connected) neural network is chosen. This should be motivated/explained in the text.

- Fig. 1d: the inset is so tiny that it's impossible for me to figure out what is going on there with the different colors.

- Fig. 2: it is very hard to see any difference between the legitimate and the adversarial density matrix -- maybe it would be more useful to plot the difference?

- Fig. 2 caption: 'different layers of kz ' - I'm assuming this means different values of kz ? Why the term layers? Why the representation on this pizza-graph instead of a line or scatter plot? This is obviously a matter of taste, but it definitely took me a while to understand what is actually shown here.

- Fig. 2g: for how many adversarial samples are these results?

List of major changes in manuscript (NCOMMS-21-45179) (marked in red in the main text and the Supplementary Materials):

1. Following the reviewers' suggestions, in addition to studying the adversarial perturbations that are carefully designed, we have added calculations on the influence of experimental noises in studying topological phases with the machine learning approach.
2. We have replaced several subfigures in Fig.1 and Fig.2, and revised the captions accordingly. The original Fig.1d (which shows the classifier's training process on numerically generated data) has been moved to the Supplementary Materials.
3. We have added the definitions of the topological invariant in the main text. We have moved the discretization scheme to measure the Hopf index with experimental data to the "Methods" part.
4. We have added a section about "The experimental noise simulation and random dropping trials" in the "Methods" part.
5. We have revised the abstract, the introduction and the discussion sections.
6. We have added a section about "Adversarial Training" in the Supplementary Materials to study the defense for both carefully designed and experimentally inevitable perturbations.
7. We have added other necessary revisions throughout the whole manuscript to improve the presentation and address the reviewers' comments/suggestions.

Response to Reviewer #1:

We thank the reviewer for judging our experimental results "of very high quality" and for pointing out that "The topic is timely and of clear relevance for ML community and also for people working in the characterization of quantum systems and quantum technologies." We also appreciate his/her valuable suggestions which have helped us improve the paper. We take these comments and suggestions very seriously and have done a substantial amount of new calculations accordingly to show that: (a) there exists a trade-off between data efficiency and robustness to experimental noises; (b) adversarial training can enhance the robustness of the phase classifiers against experimental adversarial perturbations. These newly added results improve the manuscript significantly and we believe it is now suitable for publishing in Nature Communications. The detailed response to the reviewer's comments (point by point) is provided below.

Comment 1 of Reviewer #1: "Review of Experimental demonstration of adversarial examples in learning topological phases: In the paper by Zhang et al., Experimental demonstration of adversarial examples in learning topological phases, the authors show that state of the art CNN models, capable of classifying experimental data from an NV centre in diamond, are strongly affected by adversarial examples. Adversarial examples are carefully crafted forms of noise obtained from the fully trained model that are superimposed to the experimental data. They trick the model into failing the predictions, as reported several times in literature, but require full access to the trained model and need to counterfeit the experimental data.

The topic is timely and of clear relevance for the ML community and also for people working in the characterization of quantum systems and quantum technologies. I enjoyed reading the article, and the main

ideas are mostly, well presented, and scientific experimental results are of very high quality, in line with previous results from the same authors. In the absence of adversarial examples, their machine learning models show state of the art performance. Although, the paper could and should be improved to highlight their main claims.”

Authors’ response: We thank the reviewer for his/her accurate summary of the paper. We agree with the reviewer that “The topic is timely and of clear relevance for the ML community and also for people working in the characterization of quantum systems and quantum technologies”. In particular, we appreciate the reviewer’s evaluation that “scientific experimental results are of very high quality”. Indeed, this experiment requires very high fidelity for all the momentum points and we have spent great efforts in pushing the average fidelity to 99.6%. This is crucial for our experimental demonstration of adversarial examples in learning topological phases: (a) In the original manuscript, we showcased that adding only a small amount of carefully designed perturbations on these accurately implemented topological phases can mislead the classifier. (b) In the revised manuscript, we have carried out additional calculations to show that even we made a great effort on improving the fidelity, the inevitable experimental noises will be more likely to become adversarial perturbations when the neural net-based classifiers present more advantages on data efficiency. These results illustrate that the vulnerability to tiny perturbations is the intrinsic properties of neural network-based classifiers on classifying phases of matter, regardless of how accurately the experiment is carried out in the lab.

Comment 2 of Reviewer #1: “From the scientific point of view, I have these major doubts:

1. If they claim that ML cannot be trusted in every scenario, I agree with the authors, but this would not be a novel result. The central claim and its importance need to be highlighted in order to assess properly the impact of their work.”

Authors’ response: We agree with the reviewer that adversarial examples have been widely studied in the classical machine learning literature. However, we stress that this is the first *experimental* demonstration of adversarial examples in the context of machine learning phases of matter. In particular, we demonstrated for the first time that: (a) adversarial examples indeed can be implemented in experiment despite inevitable experimental imperfections. We mention that the existence of numerical adversarial examples *does not* guarantee that we can implement them in real experiments—the experimental noises may wash out the tiny carefully-crafted adversarial perturbations; (b) the topological properties, such as the Hopf index and the linking structures, remain unchanged for experimentally implemented adversarial examples. This has *no* analog in classical adversarial learning for daily-life images, due to the absence of a sharply defined “topological invariant” and lack of tunable parameters for different phases. Thus, in our opinion, the novelty of this work is justified.

We thank the reviewer for the suggestion of highlighting the central claim and its importance. In the revised manuscript, we carried out new calculations and highlighted that adversarial perturbations could be a real threat for learning different topological phases from experimental data (see the main text and the following replies for details).

Comment 3 of Reviewer #1: “2. They show vulnerability against adversarial examples, a very well-known problem, for a specific case in physics. But this requires access to the model, and if we train again on different data points, the same adversarial examples might not work. So it is not clear how this would impact the work in the lab of scientists characterizing quantum systems without being attacked and having their experimental data tampered with. ”

Authors’ response: We thank the reviewer for raising this important point. First, we would like to clarify that the generation of adversarial examples may not require the access to the model due to the transferability property of adversarial examples: an adversarial example produced to deceive one specific learning model can deceive another different model, even if their architectures differ greatly or they are trained on different sets of training data (see for example: Szegedy *et al.*, Intriguing properties of neural networks, in Second International Conference on Learning Representations (ICLR, Banff, Canada, 2014); Goodfellow *et al.*, Explaining and harnessing adversarial examples, in Third International Conference on Learning Representations (ICLR, San Diego, 2015); Papernot *et al.*, Practical black-box attacks against machine learning, in Proceedings of the 2017 ACM on Asia Conference on Computer and Communications Security, ASIA CCS’ 17 (ACM, New York, 2017), pp. 506–519.).

This transferability property is also preserved in the case of adversarial examples in learning topological phases: we have tested a continuous adversarial example with $h = 3.2$, which is generated by the CNN classifier presented in the main text, by an additional CNN classifier trained with different dataset and a new deep neural network (DNN) with a fully-connected structure. This example has average fidelity 0.978, which is a little decreased comparing to the continuous example in the main text (0.989). However, it can successfully mislead all three classifiers, even under the condition that there is no access to the information of the other two newly added classifiers. The result is shown in the following table. The uncertainty is measured by 100 training processes with different random seeds.

Input (Model)	$P(\chi = 0)$	$P(\chi = 1)$	$P(\chi = -2)$
Leg. (original CNN)	1	0	0
Adv. (original CNN)	0	1	0
Adv. (CNN with new dataset)	0.1395 ± 0.0207	0.8603 ± 0.0207	0.0001 ± 0
Adv. (new DNN)	0.3473 ± 0.0312	0.6490 ± 0.0312	0.0037 ± 0.0003

Second, in our revised manuscript, we have carried out additional calculations and shown that even when the experimental data are not tempered with, the inevitable experimental noises may behave as adversarial perturbations when the CNN classifier attempts to be more data efficient (see the main text for details). We illustrate the classification confidences for theoretical data and data with simulated noises at different drop ratios in Fig. 1c and Fig.1d, respectively. The results show that the experimental noises are more likely to become adversarial perturbations when the dropping rate of the input data increases, and for the testing samples that are close to the phase transition points, this problem occurs more seriously. In Fig.2a we demonstrated this with experimental data. These new results show clear evidence that adversarial examples might be a real threat when using machine learning tools to classify topological phases from experimental data with inevitable noises. This also provides a useful guidance for experimental scientists in balancing the trade-off between data efficiency and classification accuracy.

Comment 4 of Reviewer #1: “3. Is the use of tomography, a highly unscalable method, for the characterization of a solid-state system something the authors would expect to be applicable in a broader scenario? How would the authors envisage the use of a similar methodology for larger systems? The number of data points would scale exponentially fast. This is a key point because if the technique is not scalable is never going to be applied in practical cases.”

Authors’ response: We thank the reviewer for raising this important concern. We agree with the reviewer

that full tomography is not a scalable method. However, in this work we are actually simulating the Hopf insulator in the momentum space and we only need to tomography single-qubit states at different momentum points. As a result, the complexity only scales linearly with the number of momentum points and the scalability is not a problem here. We mention that the simulation of the Hopf insulator in real space is extremely challenging due to complicated spin-dependent hopping terms, and to date no experiment has achieved this despite a couple of theoretical proposals (see for example: D. -L. Deng *et al.*, Probe koints and Hopf insulators with ultracold atoms, in *Chin. Phys. Lett.* 35, 013701 (2018); Thomas Schuster *et al.*, Floquet Hopf Insulators, in *Phys. Rev. Lett.* 123, 266803 (2019); Thomas Schuster *et al.*, Realizing Hopf Insulators in Dipolar Spin Systems, in *Phys. Rev. Lett.* 127, 015301 (2021)). In the future, if Hopf insulator is experimentally realized in real space with a very large system size, it is then indeed impractical to tomography the full quantum many-body states, as correctly pointed out by the reviewer. Yet, we may use other input data that can be easily obtained in experiment, such as time-of-flight images or STM images, to train the classifier and study their corresponding adversarial examples. In fact, in the theoretical paper of Ref. [30], we have done extensive numerical simulations to study adversarial examples based on local magnetizations or time-of-flight images, which can be readily measured in real experiment.

In short, we agree with the reviewer that full tomography is not scalable. However, this problem can be easily circumvented by using different input data that can be measured in experiment directly. Here, we choose the single-qubit states as input data simply because we are working in the momentum space and such input data can be readily measured in our experiment.

Comment 5 of Reviewer #1: “While I like the article and enjoyed reading it, I am not yet convinced that the results, as they are now, deserve publication in a high-impact journal such as Nature Communications. And I would be happy to be convinced that instead, they deserve publication in NC.”

Authors’ response: We are very happy to know that the reviewer “like the article and enjoyed reading it”. Based on the reviewer’s suggestions/comments, we have carried out a substantial amount of new calculations and improved the paper significantly. We hope this revised manuscript can satisfy the reviewer and convince him/her to recommend its publication in NC.

Comment 6 of Reviewer #1: “To properly assess the impact of this work, we need to ask the question, are adversarial examples a real threat for solid-state system characterization? It is unclear to me, in the current version of the manuscript, whether this is going to be the case or not. Adversarial examples are a clear threat for example, for self-driving cars, where one could place a carefully crafted image printed on a board and placed on the street to make the car crash.

But in this specific study, what kind of scenario the authors are envisaging? Do we expect somebody to steal the model used in the characterization process of the solid-state system and carefully falsify the experimental data to make it look like a system is in a phase rather than in another one? The authors should clarify in what circumstances or scenarios the adversarial examples would be used. A random noise will not have the same effects as adversarial examples, and the literature in ML is full of examples where very small perturbations, if carefully crafted, can heavily affect the performance of ML algorithms. In general, I agree with the authors we cannot trust CNN models to predict effectively against adversarial examples. This is a topic of research and an important one. Of course, if the authors provided a solution to the problem then it would be a different story, but in this case, I would be happy to have a convincing practical case where this problem would be of impact on scientific research.”

Authors’ response: We thank the reviewer for agreeing that adversarial examples “is a topic of research and an important one”. His/her concerns and questions on the motivation of studying the adversarial examples in learning phases and experimentally implementing adversarial examples have guided us to carry out new calculations and improve the manuscript significantly.

We agree with the reviewer that, while the adversarial examples are clear threats for daily-life applications, such as self-driving cars and face recognition, their consequences in the case of learning topological phases may not be as obvious and clear as in these applications. However, we argue that the study of adversarial examples in learning topological phases, or more broadly machine learning applications in physics, is of both fundamental and practical importance due to the following reasons:

- (i) As mentioned above, the existence of numerical adversarial examples does not guarantee that we can implement them in real experiments due to inevitable experimental imperfections. Thus, from the fundamental research perspective, it is important and necessary to carry out an experiment to demonstrate that adversarial examples can indeed be engineered in the laboratory.
- (ii) Such a study could help deepen our understanding for the limitations and advantages of using machine learning techniques to identify phases of matter. Our results clearly reveal that current machine learning approaches may not fully capture the underlying physical principles.
- (iii) In the future, topological materials may have applications in information processing. Sensitive information might be encoded with different topological states, and in such a scenario adversarial examples of topological states would be a threat in practice.

In this revised manuscript, we have carried out new calculations to show that even when the experimental data are not tempered with, the inevitable experimental noises may become adversarial perturbations when a large portion of the data are dropped out or inaccessible (see Fig. 1c, Fig. 1d, and Fig. 2a in the main text for details). More specifically, in Fig. 1c, we show that the classifier can correctly classify the clean data of topological trivial phase ($h = 3.2$, $\chi = 0$) with more than 80% of the data dropped, but in this case, more than 50% of simulated experimental noise will behave as adversarial perturbations. This problem becomes more serious when the testing samples are closer to the phase transition point, as shown in Fig. 1d. We also showcase this problem on the real experimental data: as shown in Fig. 2a, we randomly drop the experimentally implemented density matrices of the legitimate sample with $h = 3.2$. We find that the the random experimental noises mislead the classifier more and more seriously as the ratio of dropped data increases, and finally becomes the adversarial perturbation when 60% of data are dropped. These new results answer the reviewer’s question about whether “A random noise will have the same effects as adversarial example”, indicating that there is a trade-off between the data efficiency and robustness to the tiny experimental noises. They show clearly that adversarial examples might be a real threat when using machine learning tools to classify topological phases from experimental data with inevitable noises. A balance of the trade-off between data efficiency and classification accuracy should be maintained in practice.

In addition, following the reviewer’s suggestion that “if the authors provided a solution to the problem then it would be a different story”, we have studied a useful defense strategy against adversarial perturbations, namely a variation of the adversarial training approach. The essential idea is to retrain the classifier by legitimate and adversarial data alternatively. To demonstrate the effectiveness of this approach in learning topological phases with experimental data, we modify the adversarial training algorithm and make it also takes the randomly dropped data into account. As shown in Supplementary figure 9a, the classifier after adversarial training can correctly classify the sample with more than 90% of data dropped, and the ratio of adversarial perturbations in the experimental noises increases much slower. This shows that adversarial

training can indeed enhance the robustness of classifiers against both carefully designed and experimental noises in real experiment. In the revised manuscript, we mentioned these new results in the discussion section and added the details in the Supplementary Materials Sec. 8.

In summary, we greatly appreciate the reviewer's valuable suggestions/comments, which have guided us to carry out a substantial amount of new calculations and improve the manuscript significantly. We believe that this work is important for the rapidly growing interdisciplinary field of machine learning phases of matter. We hope this significantly strengthened manuscript will satisfy the reviewer and convince him/her to recommend its publication in Nature Communications.

Response to Reviewer #2:

We sincerely thank the reviewer for his/her careful reading of our manuscript and judging our experiment "fidelities are very high and convincing, and the implementation of the adversarial attack is done with very high precision". We also appreciate his/her valuable suggestions, In addressing these suggestions/comments, we have carried out a substantial amount of additional calculations to show that: (a) the inevitable experimental noises will be more likely to become adversarial perturbations when the neural network-based classifiers present more advantages on data efficiency. (b) adversarial training can enhance the robustness of the phase classifiers against adversarial perturbations. The newly added results improve the paper significantly and we believe it is now suitable for publishing in Nature Communications. The detailed response to the reviewer's comments is provided below.

Comment 1 of Reviewer #2: "In "Experimental demonstration of adversarial examples in learning topological phases" the authors experimentally demonstrate the failure of a classification based on neural nets upon adversarial attacks. They use an NV center for the study of the dispersion of a 3D Hopf insulator. Based on the reconstructed density matrices they first train a CNN to predict the Hopf index, with great success. Then they perform an optimization problem to devise local perturbations that maximize the loss function and hence fool the neural net. They can implement these adversarial attacks in the experiment, and they find that the phase classification has become useless indeed. It follows that the neural net has not learned the correct physical property based on topology. It is unclear what was learned instead. "

Authors' response: We thank the reviewer for the accurate summary of the main results of our paper.

Comment 2 of Reviewer #2: "From the theoretical perspective, the conclusions in this work are not at all surprising. It is well-known that a naive classifier fails to adequately capture topological information. But the field has moved on; there are now several papers addressing topological phases with machine learning, not only the ref 15-17 in the paper. We also know what to do. Essentially, we need to know about the behavior of the curvature around high symmetry points such as $(0,0,0)$, $(0,0,\pi)$ etc. (so you should not need all k-points, you can reduce the input dimension!). It is possible to correctly classify certain topological phases based on this curvature. The curvature function typically involves derivatives of the phase. The reduced density matrices, which serve as the input here, contain very little information about the phase. So one can equally say the problem lies with the humanly chosen input than with the neural network."

Authors' response: We thank the reviewer for bringing up the latest progress. Following his/her suggestions, we have added some latest researches on this field (Ref [14]: P. Mognini *et al.*, A supervised learning algorithm for interacting topological insulators based on local curvature, in *SciPost Phys.*, 11, 73(2021); Ref [21]: Niklas Käming *et al.*, Unsupervised machine learning of quantum phase transitions from experimental data, in *Mach. learn.: Sci. and Technol.*, 2,035037(2021)). However, due to the limited number of references required by Nature Communications, we are not able to cite all papers in this direction.

We agree with the reviewer that for some topological phases, we only need to know about the behavior of the curvature around high symmetry points. However, this requires prior knowledge about the properties of the characterizing topological invariants and may not be generally applicable to different topological phases without certain symmetries. One of the major motivations of using neural networks to identify topological phases is that we may be able to classify different topological phases without knowing the details of how to calculate the characterizing topological invariants. In our work, instead of learning the topological phase with curvature, we aim to detect the Hopf index directly from experimental raw data. The neural network does not require to know the precise calculation for different topological indices, so is especially helpful in the cases when the topological characters are less understood. Even though the reduced density matrices do not contain the whole information of the phase, the trained CNN model can still classify the legitimate samples for different Hopf indices with high confidence. In addition, we also clarify that the existence of adversarial examples is widely believed to be intrinsic for learning from high-dimensional data, independent of how we choose the input data. This is reminiscent of the “no free lunch” theorem—there exists an intrinsic tension between adversarial robustness and generalization accuracy (see the newly added Refs. [68-70]). In the revised manuscript, we have clarified this important point in the discussion section.

Comment 3 of Reviewer #2: “Experimentally the results are convincing. The reconstruction of the density matrices based on quantum state tomography and maximum likelihood estimation works well for this system, and are of course a convenient input for the neural net. The reported fidelities are very high and convincing, and the implementation of the adversarial attack is done with very high precision too. The text is very well written and can be understood by non-experts. Although I am in general all in favor of publishing a negative result, I must say that this negative result here is totally expected. The sentence “How to exploit the experimental data to make the classifier better learn physical principles, and how to protect the neural-network based classifiers in real experiments from adversarial perturbations, are interesting and important problems worth further investigation” in the conclusion is however below my expectations: I would have expected more on this, given that the field has moved on and that ideas of classifying topology do exist in the literature. ”

Authors' response: We thank the reviewer for judging our experiment “fidelities are very high and convincing, and the implementation of the adversarial attack is done with very high precision”. Indeed, in this manuscript, we report the first proof-of-principle experimental demonstration of adversarial examples in learning topological phases. Our results uncover the potential challenges in applying machine learning approaches to study topological phases in real experiment for the first time. This experiment requires very high fidelity for all the momentum points and we have spent great efforts in pushing the average fidelity to 99.6%. This is important for our experimental demonstration of adversarial examples in learning topological phases: (a) In the original manuscript, we showcased that adding only a small amount of carefully designed perturbations on these accurately implemented topological phases can mislead the classifier. (b) In the revised manuscript, we have carried out additional calculations to show that even we made a great effort

on improving the fidelity, the inevitable experimental noises will still be more likely to become adversarial perturbations when the neural network-based classifiers present more advantages on data efficiency. These results illustrate that the vulnerability to tiny perturbations is an intrinsic property of neural network-based classifiers on classifying phases of matter, regardless of how accurately the experiment is carried out in the lab. We stress that, far from discouraging continued studies on machine learning phases of matter, our aim of demonstrating adversarial examples in experiment is to deepen our understanding of such perturbations so that we can develop better countermeasures, and to inspire more investigations along this rapidly growing interdisciplinary direction.

In this revised manuscript, we have carried out new calculations to show that the inevitable experimental noises may become adversarial perturbations when a large portion of the data are dropped out or inaccessible (see Fig. 1c, Fig. 1d, and Fig. 2a in the main text for details). In particular, in Fig. 1c we show that the classifier can correctly classify the clean data of topological trivial phase ($h = 3.2$, $\chi = 0$) with more than 80% of the data dropped. However, with this drop ratio, more than 50% of simulated experimental noise will behave as adversarial perturbations. This problem becomes even more serious near the phase transition point, as shown in Fig. 1d. We also showcase this problem on the real experimental data (see Fig. 2a). We find that the random experimental noises mislead the classifier more and more seriously as the ratio of dropped data increases, and finally becomes the adversarial perturbation when 60% of data are dropped. These newly added results imply that there is a trade-off between the data efficiency and robustness to the tiny experimental noises. As a result, a balance of the trade-off between data efficiency and classification accuracy should be maintained in practice.

We thank the reviewer for raising the comment “is however below my expectations: I would expected more on this,...”. This useful comment has motivated us to study an effective defense strategy against adversarial perturbations, namely a variation of adversarial training. The essential idea is to retrain the classifier with both legitimate and adversarial data. To demonstrate the effectiveness of this approach in learning topological phases with experimental data, we modify the conventional adversarial training algorithm and make it also take the randomly dropped data into account. We find that after the adversarial training, the phases classifier will restore its correctness for experimentally implemented adversarial examples, and can also correctly classify these samples with more than 90% of data dropped. These results clearly show that adversarial training can indeed enhance the robustness of the classifiers against both carefully designed and experimental noises in real experiment. In this revised manuscript, we have mentioned these new results in the discussion section of the main text and added the details in the Supplementary Materials Sec. 8.

In summary, we greatly appreciate the reviewer’s valuable suggestions/comments, which have motivated us to carry out a substantial amount of new calculations. Based on his/her report, we improved the manuscript significantly and clarified several important points. We hope this substantially strengthened manuscript will satisfy the reviewer and convince him/her to recommend its publication in Nature Communications.

Response to reviewer #3:

We sincerely thank the reviewer for his/her careful reading of our manuscript and appreciate his/her valuable suggestions/comments, which have helped us to improve the paper significantly. In addressing these suggestions/comments, we have done a substantial amount of additional calculations to justify the importance and necessity of experimental demonstration of adversarial examples. Based on his/her reports, we have also improved the presentation of the manuscript and clarified some important points. Now, the paper is

notably strengthened and we believe it is suitable for Nature Communications. The detailed response to the reviewer’s comments is provided below.

Comment 1 of Reviewer #3: “In their paper ‘experimental demonstration of adversarial examples in learning topological phases’, the authors study the Hopf model using a convolutional neural network. They employ both numerical simulation as well as experiments on an NV center to generate the quantum state of the system for different momenta and values of the Hamiltonian parameter h . They use different methods to find adversarial examples which will most confuse the neural network, and implement those examples both numerically as experimentally, and show that the neural network can indeed be deceived by those examples. The general topic of adversarial examples is important when using machine learning tools for scientific insights, and I believe the authors made some progress in the present work. However, several things in the manuscript are unclear to me and therefore hard to judge.”

Authors’ response: We thank the reviewer for the accurate summary of the main results of this work and for pointing out that “The general topic of adversarial examples is important when using machine learning tools for scientific insights”. We fully agree with the reviewer on this point. We also appreciate the reviewer’s positive judgement that “the authors made some progress in the present work”. Indeed, this is the first experimental demonstration of adversarial examples in the rapidly growing direction of machine learning phases of matter. We are very excited about the completion of this experiment and believe it will have far-reaching impact for future applications of machine learning techniques in identifying exotic phases of matter.

Comment 2 of reviewer #3:(1) “- what is a Hopf insulator – despite the paragraph on p.2 about this model, it is not clear to me why this model has topological phases of matter, (2)or how this is related to the NV center described earlier,or how one arrives at the Hopf index. Since this seems to be (?) what the network is trained to predict, it should be explained in more detail, and its relation to the density matrix should be clarified. ”

Authors’ response: We thank the reviewer for raising this point, which are very helpful for us to improve the presentation.

(1) For free fermions, most 3D topological insulators have to be protected by some other symmetries such as time-reversal, particlehole, or chiral symmetry, and the U(1) charge conservation symmetry (see e.g., Refs.[50,51]). A peculiar exception occurs when the Hamiltonian has only two effective bands, where the Hopf insulating phases can exist due to the nontrivial Hopf map. These Hopf insulator phases have no symmetry other than the prerequisite U(1) charge conservation. Hopf insulators are characterized by an integer topological invariant—the Hopf invariant. With open boundary condition, they have topologically protected surface states (see Refs.[48,49]). We mention that the realization of Hopf insulator in real space is extremely challenging due to the complicated spin-dependent hopping terms, and to date no experiment has achieved this despite several theoretical proposals (see e.g., Chin. Phys. Lett. 35, 013701 (2018); Phys. Rev. Lett. 123, 266803 (2019); Phys. Rev. Lett. 127, 015301 (2021)). In this work, we simulated the single-particle Hopf Hamiltonian in the momentum space, which is technically much easier in the laboratory.

(2) We have added the definition of the Hopf index χ in the revised manuscript. The Hamiltonian of the Hopf insulator in momentum space can be written as :

$$H_{\text{TI}} = \sum_{\mathbf{k} \in \text{BZ}} \Psi_{\mathbf{k}}^\dagger H_{\mathbf{k}} \Psi_{\mathbf{k}} = \sum_{\mathbf{k}} \Psi_{\mathbf{k}}^\dagger u_{\mathbf{k}} \cdot \boldsymbol{\sigma} \Psi_{\mathbf{k}}, \quad (1)$$

where $\mathbf{u}_{\mathbf{k}} = (u_x, u_y, u_z)$, and $\boldsymbol{\sigma} = (\sigma_x, \sigma_y, \sigma_z)$ are Pauli matrices. In the matrix form, the Hamiltonian reads:

$$H_{\mathbf{k}} = \begin{pmatrix} 0 & u_x - iu_y \\ u_x + iu_y & -u_z \end{pmatrix}. \quad (2)$$

The effective Hamiltonian of the NV center in the rotating frame can be written as:

$$H_{\text{eff}} = \begin{pmatrix} 0 & |\Omega|e^{i\varphi} \\ |\Omega|e^{-i\varphi} & -\Delta\omega \end{pmatrix}. \quad (3)$$

Thus, if we carefully design the pulse sequence and choose appropriate microwave amplitude $\Omega(t)$, phase φ , and frequency detuning $\Delta\omega$ (so that $\Delta\omega/|\Omega| = u_z/\sqrt{u_x^2 + u_y^2}$, $\varphi = -\arctan(u_y/u_x)$), H_{eff} reduces to $H_{\mathbf{k}}$ (up to an unimportant overall normalization) and we can simulate the Hopf Hamiltonian in the momentum space with our NV center platform.

The reviewer's questions reflect that we probably did not explain clearly enough about what are Hopf insulators and how to simulate them with NV centers. In the revised manuscript, we have added more discussions on these points and improved the presentation accordingly.

Comment 3 of Reviewer #3: “- what is predicted by the neural network? This is a crucial question that should be very clear, but is not to me from the present text.”

Authors' response: We thank the reviewer for raising this point. In our experiment, the output layer of the neural network contains three neurons, returning the probabilities of the Hopf index being 0, 1, and -2, respectively. The predicted Hopf index for each sample is the one that has the maximal probability. The reviewer's question reflects that we probably did not explain this clearly enough. In this revised manuscript, we have clarified this point and improved the overall presentation.

Comment 4 of Reviewer #3: “- (1) Fig. 2h: it would be very useful if the reader knew how the Hopf index is obtained from the density matrix. (2) In particular, how the density matrices for different momenta and different values of h are connected. (3) Is it possible that a reordering in momentum space (possible obtained through convolution and max pooling in the neural network?) leads to the wrong classification? (4) As the authors themselves write “This result is at variance with the physical intuition that Hopf insulators are robust to local perturbations due to their topological nature [45, 46], indicating that the neural network based classifier does not fully captured the underlying topological characteristics [27].” – this makes it even more important to explain to the reader why the model should be robust to local perturbations, and what it actually is that they ask their network to predict. Without this information the results are very hard to appreciate. Given the next paragraph in the text, it is even less clear to me what they were classifying before.”

Authors' response: We thank the reviewer for these constructive suggestions, which are very helpful in improving the manuscript.

(1) We have added the detailed method for calculating χ in the main text and the “Methods” part. The Hopf index χ is defined as:

$$\chi = - \int_{\text{BZ}} \mathbf{F} \cdot \mathbf{A} d^3\mathbf{k}, \quad (4)$$

\mathbf{F} is the Berry curvature, and the discretized version of \mathbf{F} is defined as:

$$F_{\mu}(\mathbf{k}_{\mathbf{J}}) = \frac{i}{2\pi} \epsilon_{\mu\nu\tau} \ln U_{\nu}(\mathbf{k}_{\mathbf{J}}) \ln U_{\tau}(\mathbf{k}_{\mathbf{J}+\hat{\nu}}), \quad (5)$$

where

$$U_\nu(\mathbf{k}_J) = \frac{\langle \psi(\mathbf{k}_J) | \psi(\mathbf{k}_{J+\hat{\nu}}) \rangle}{|\langle \psi(\mathbf{k}_J) | \psi(\mathbf{k}_{J+\hat{\nu}}) \rangle|}, \quad (6)$$

In experiment, we perform state tomography at $10 \times 10 \times 10$ momentum points to obtain $|\psi(\mathbf{k}_J)\rangle$. Then one can obtain $U_\nu(\mathbf{k}_J)$ by Eq. (6), and subsequently $F_\mu(\mathbf{k}_J)$ through Eq. (5). After obtaining $F_\mu(\mathbf{k}_J)$, the Berry connection \mathbf{A} can be obtained by Fourier transforming $\nabla \times \mathbf{A} = \mathbf{F}$ with the Coulomb gauge $\nabla \cdot \mathbf{A} = 0$. Finally, we sum over the $10 \times 10 \times 10$ points on the momentum grid to obtain the Hopf index χ .

(2) Since the Hamiltonian of the topological insulator in the momentum space is described by $u_{\mathbf{k}} = \{u_x, u_y, u_z\}$, $u_{\mathbf{k}}$ is connected with h by:

$$u_x = 2(\sin k_x \sin k_z + C_{\mathbf{k}} \sin k_y), \quad (7)$$

$$u_y = 2(C_{\mathbf{k}} \sin k_x - \sin k_y \sin k_z), \quad (8)$$

$$u_z = \sin^2 k_x + \sin^2 k_y - \sin^2 k_z - C_{\mathbf{k}}^2, \quad (9)$$

$$C_{\mathbf{k}} \equiv \cos k_x + \cos k_y + \cos k_z + h. \quad (10)$$

Therefore, the ground states $|\psi(\mathbf{k}_J)\rangle$ depend on the parameter h .

(3) In general, a reordering in momentum space is a global operation, thus could change the topological invariant. Since this may completely change the data sample, whether the prediction of the classifier for the reordered sample is correct or not becomes not a well-defined question in this scenario.

(4) In general, a topological invariant characterizes the global properties of the system and should be robust to any local perturbations. In our case, the neural network is supposed to learn the map from data samples to topological invariant. More concretely, the neural network predicts the probabilities of the Hopf index of the input data. The result $P(\chi = 0)$, $P(\chi = 1)$ and $P(\chi = -2)$ is the probability of the Hopf index being 0, 1, -2, respectively. However, in our experiment we have shown that adding a tiny amount of local perturbations to a legitimate sample could deceive the neural network. This indicates that the neural network does not really learn the underlining topological invariant and associated physics.

Comment 5 of Reviewer #3:(1) “- I don’t really understand why it’s important to experimentally engineer one particular adversarial example. Isn’t it rather the other way round, that we should make sure that experimental imperfections don’t lead to (undesired!) adversarial examples? (2) This leads to the question of how fine-tuned these adversarial examples are. I’m assuming that this single qubit system is experimentally very well controlled, also given the fidelities in preparing a given quantum state demonstrated in the paper. In particular, given the fidelity of 99.23% between the numerical and experimental adversarial example, it doesn’t seem that there are too many experimental errors in play here. It’s unclear what happens with a slightly less fine-tuned adversarial example – does it still get mis-classified? (3) Or do we only get this strong effect if we choose one very very specific state? In my opinion, this makes a big difference for the importance of this problem. ”

Authors’ response: We thank the reviewer for raising these penetrating questions, which have led us to carry out a substantial amount of additional calculations and clarify some important point. In the following, we address the reviewer’s questions, one by one.

(1) The importance and necessity of experimental demonstration of adversarial examples can be justified from the following reasons: a) although the existence of adversarial examples in learning topological phases has been theoretically predicted in Ref. [30] with extensive numerical simulations, it is not guaranteed that

we can easily implement them in real experiments due to inevitable experimental imperfections. The experimental noises may wash out the tiny carefully-crafted adversarial perturbations. Thus, an experimental demonstration of adversarial examples is still necessary and important; b) the study of experimental adversarial examples would help us (quote the reviewer) “make sure that experimental imperfections don’t lead to (undesired!) adversarial examples”. In fact, this is also one of our motivations for this work. In the revised manuscript, we have carried out new calculations and shown that the natural inevitable experimental noises may also behave as adversarial perturbations when the classifier attempts to be more data efficient (see Fig. 1c, Fig. 1d) and Fig. 2a in the main text for details). These new results imply that, in order to prevent experimental imperfections from becoming adversarial examples, it is important that the data dropping ratio should not be too large.

(2) This question concerns the robustness of the adversarial examples to experimental noises. In general, this depends on the preset threshold fidelity between the adversarial and legitimate samples. Typically, this fidelity is set to be a very high value so as to assure that the adversarial examples do not differ too much from the legitimate samples. As a result, in order to implement adversarial examples the experimental noises should be very small (otherwise they may wash out the tiny carefully-crafted adversarial perturbations). This is why we have spent great efforts to push the average fidelity to 99.6% (which is nontrivial for the NV platform in quantum simulations) in our experiment.

(3) We clarify that the adversarial examples are not specific to a particular legitimate sample. In fact, for almost all samples, one can find their corresponding adversarial samples through various optimization strategies (see Ref. [30]). Theoretically, the existence of adversarial examples is a fundamental feature of machine learning in high-dimensional spaces rooted in the concentration of measure phenomenon. In addition, we also mention that in order to make a legitimate sample to be adversarial examples, the desired adversarial perturbation is not unique. As shown in our experiment, we can add continuous or discrete perturbations to obtain adversarial examples from legitimate samples.

Comment 6 of Reviewer #3: “- the importance and difference between continuous and discrete perturbations is not clear to me.”

Authors’ response: In this paper, the continuous perturbation is generated by adding the perturbation on all the density matrices, while the discrete perturbations are only added to seven k points. We aim to show that the different types of adversarial perturbations can be obtained by different approaches. The Projected Gradient Descent (PGD) method and Momentum Iterative Method (MIM) are applied to generate continuous perturbations and the Differential Evolution Algorithm (DEA) is applied to generate discrete perturbations. Our results show that adversarial examples obtained by adding both continuous and discrete perturbations can be implemented in experiment, regardless the inevitable experimental imperfections.

Comment 7 of Reviewer #3: “- Fig. 3: to really appreciate what’s happening here and how surprising it is, it would be tremendously helpful to also show the corresponding images for an example from the phase (?? Or value of χ ?) AS WHICH the neural network wrongly classifies the adversarial example. This would greatly help to understand how close this adversarial example is to the ‘right’ vs ‘wrong’ phase. Another way to showcase this would be to calculate fidelities to all possible states from the ‘wrong’ phase, because in the end the question is whether these fidelities are low compared to the confidence of the network in its wrong classification. This very crucial piece of information is not clear to me from the paper.”

Authors' response: We thank the reviewer for this helpful suggestions. We have replotted Fig. 3 and added explicitly the classification result of the legitimate example, continuous adversarial examples and discrete adversarial examples on the figure, so it is easier for readers to reach the conclusion.

Comment 8 of Reviewer #3: “Minor comments:

- Fig. 1a is a bit confusing – unclear what’s supposed to be 3D and what not – where is the vacancy?
- Fig. 1 caption: “indexes” (indices?)
- Fig. 1c: it is not clear to me why a convolutional (as opposed to a fully connected) neural network is chosen. This should be motivated/explained in the text.
- Fig. 1d: the inset is so tiny that it’s impossible for me to figure out what is going on there with the different colors. ”
- Fig. 2: it is very hard to see any difference between the legitimate and the adversarial density matrix – maybe it would be more useful to plot the difference?
- Fig. 2 caption: ‘different layers of kz ’ - I’m assuming this means different values of kz ? Why the term layers? Why the representation on this pizza-graph instead of a line or scatter plot? This is obviously a matter of taste, but it definitely took me a while to understand what is actually shown here.
- Fig. 2g: for how many adversarial samples are these results? ”

Authors' response: We thank the reviewer for his/her very careful reading of our manuscript and for raising these minor comments, which are very helpful for us to improve the manuscript. In the following, we address these comments, one by one:

- We have replotted the Fig. 1a, and highlighted the vacancy (V) by blue color. Now the 3D structure of the NV center is much clearer.
- The typos in the caption are corrected.
- There are two reasons for choosing the convolutional neural network (CNN). First, previous works have demonstrated that CNN has an excellent performance in identifying topological phases and is the most widely used network in the related literature (see for example: Carrasquilla and Melko, Machine Learning Phases of Matter (Nature Physics, 2017); Bohrdt *et al.*, Classifying snapshots of the doped Hubbard model with machine learning (Nature Physics, 2019)); Second, CNN naturally features a translational symmetry, which is more convenient for learning phases in the momentum space. In the revised manuscript, we have clarified this important point in the main text “Machine learning of topological phases” part.
- We have replaced Fig. 1d with a new figure on “the ratio of adversarial perturbations around the phase transition point”. We have moved the original Fig. 1d to the Supplementary Materials Sec. 5. Now, the size of this figure is substantially enlarged.
- A key feature of adversarial examples is that: although they differ from their corresponding legitimate samples by only a tiny amount of noises (even imperceptible to human eyes), they can successfully deceive the classifiers. In Fig. 2a and 2b (Fig. 2b and Fig. 2c in the revised manuscript), we aim to show this feature with experimentally implemented legitimate and adversarial examples. Thus, we think it is better to plot these samples directly (rather than their difference), so as to stress the similarity between the legitimate and adversarial samples.

- We have replaced “different layers of kz ” with “different values of kz ” in the caption of Fig. 2. We use the representation of pizza-graph because the momentum space k is periodic.
- In Fig. 2g (Fig. 2f in the revised manuscript), there is only one adversarial sample for $h = 0.5$, $h = 2$, and $h = 3.2$, respectively. For the single adversarial sample, the CNN classifier outputs the classification confidences of the Hopf index.

In summary, we greatly appreciate the reviewer’s careful reading of the manuscript. Based on his/her reports, we have done a substantial amount of additional calculations and improved the presentation significantly. We have carefully addressed all the comments/suggestions raised by the reviewer. We hope this significantly enhanced manuscript will satisfy the reviewer and convince him/her to recommend publication of this work in Nature Communications.

REVIEWERS' COMMENTS

Reviewer #1 (Remarks to the Author):

Review of Experimental demonstration of adversarial examples in learning topological phases
Rebuttal

In the revised manuscript the authors have addressed some of my major concerns, improving substantially some of the content. I, in particular, appreciate the additional analysis on the relation between classification errors and experimental noise.

However, I believe that the overall quality of the paper needs to be further improved before it can be published in a high-impact journal such as Nature communication.

One of the major points that still needs some work is the justification of how the work is relevant to a broad scientific audience, like the one in Nature communication.

The authors also state in their response:

The results show that the experimental noises are more likely to become adversarial perturbations when the dropping rate of the input data increases, and for the testing samples that are close to the phase transition points, this problem occurs more seriously. In Fig.2a we demonstrated this with experimental data. These new results show clear evidence that adversarial examples might be a real threat when using machine learning tools to classify topological phases from experimental data with inevitable noises. This also provides useful guidance for experimental scientists in balancing the trade-off between data efficiency and classification accuracy.

While I appreciate their efforts in clarifying this point, I still cannot see how we can confuse experimental noise and adversarial attacks. The two, in principle, are very different, even if the outcome can be similar in some cases. Noise can cause misclassification, but this is different from a purposely designed perturbation of the data that triggers a misclassification in a specific way.

Furthermore, though less critical, I would suggest that the authors carefully review the use of proper scientific English in their manuscript.

I will not go into the details of the whole paper, which requires a proper review by the authors on the language used, but for example, the modification in the abstract states:

""We show that the experimental noises will be more likely to become adversarial perturbations when the neural net-based

classifiers present more advantages on data efficiency. We implement experimentally several adversarial examples which can deceive the splendid phase classifier with a notably high confidence, while keeping their physical properties unchanged.""

Not only the language needs to be improved, but also the authors should avoid, in the whole manuscript, using extremely qualitative and colloquial terms such as "splendid". Also, what are the physical properties left unchanged here, those of the system or those of the classifier?

What do we mean in the context of the abstract for more advantages? It is somewhat a vague and unclear sentence.

Other parts of the manuscript present similar problems.

Reviewer #2 (Remarks to the Author):

I have read the revised manuscript and the reply written by the authors. The main strength of the paper is that the authors have experimentally demonstrated an adversarial attack for the classification of a topological index despite a very high fidelity. They also show that experimental noise could be relevant for such attacks in certain cases. Although the concept of adversarial attacks is certainly not new, its casting in an experimental context with a careful analysis is important.

The presentation of the paper has much improved. The reply looks convincing to me.

I recommend publication of this paper.

Reviewer #3 (Remarks to the Author):

I have carefully read the response of the authors to my report as well as to the other referee's reports and the revised manuscript. The authors have extensively replied and reacted to my comments and questions. In particular, they have added a completely new analysis of the potential of experimental noises as adversarial examples, which I find very interesting and highly relevant. As more and more machine learning techniques are applied to experimental data, the influence of experimental errors/noise becomes a pressing question that needs to be addressed in order to be certain about what is being learned. The authors make an important step in this direction with their revised manuscript. I recommend publication in Nature Communications.

Minor comments/typos:

Introduction, p.1: "In this work, we find that the adversarial examples are likely to occur by experimental noises instead of carefully crafted perturbations when the classifier presents more advantages on data efficiency."

-- I find this sentence somewhat confusing: I would think adversarial examples occur equally by experimental noises and carefully crafted perturbations? And 'presents more advantages on data efficiency' is somewhat vague, could be more specified.

p.1: "We then show that, even our legitimate samples have very high fidelity (99.7% on average)," -- I think there might be a 'though' missing (even though our legitimate samples..)?

p.2: "We experimentally implemented the adversarial examples based on legitimate samples without data dropping." This sentence is not clear to me: what is happening now?

Fig.1c: why is the green line ($P(\chi^2=2)$) always zero?

Fig.1d: the y-axis label seems to appear twice.

p.5 "typical in experiments would nullify the adversarial examples. In the worst case" -- space missing after full stop.

p.6 "As a possible solution to this problem, We tried a variation of the adversarial training" (We->we)

p.6 "As a possible solution to this problem, We tried a variation of the adversarial training, which takes both carefully de- signed adversarial examples and randomly dropped data into account. While this defense strategy can indeed im- prove the robustness to carefully designed perturbations and experimental noises when a large portion of data are dropped, the classifier's performance near the phase transition points get much worse."

-- it's not entirely clear to me whether this is something that has been done/discussed in the main text before the Discussion section? If yes, I didn't realize it when reading the manuscript and the authors might want to consider to highlight it a bit more.

Response to Reviewer #1:

Comment 1 of Reviewer #1: Review of Experimental demonstration of adversarial examples in learning topological phases Rebuttal: In the revised manuscript the authors have addressed some of my major concerns, improving substantially some of the content. I, in particular, appreciate the additional analysis on the relation between classification errors and experimental noise. However, I believe that the overall quality of the paper needs to be further improved before it can be published in a high-impact journal such as Nature communication.

Authors' response: We thank the reviewer for his/her positive evaluation of our newly added calculations and results. We also greatly appreciate his/her valuable suggestions and comments, which have helped us to further improve the paper.

Comment 2 of Reviewer #1: One of the major points that still needs some work is the justification of how the work is relevant to a broad scientific audience, like the one in Nature communication. The authors also state in their response: The results show that the experimental noises are more likely to become adversarial perturbations when the dropping rate of the input data increases, and for the testing samples that are close to the phase transition points, this problem occurs more seriously. In Fig. 2a we demonstrated this with experimental data. These new results show clear evidence that adversarial examples might be a real threat when using machine learning tools to classify topological phases from experimental data with inevitable noises. This also provides useful guidance for experimental scientists in balancing the trade-off between the cost of generating experimental data and classification accuracy.

While I appreciate their efforts in clarifying this point, I still cannot see how we can confuse experimental noise and adversarial attacks. The two, in principle, are very different, even if the outcome can be similar in some cases. Noise can cause misclassification, but this is different from purposely designed perturbation of the data that triggers a misclassification in a specific way.

Authors' response: We thank the reviewer for raising this important point. We agree with the reviewer that the experimental noises and adversarial attacks are very different, in the sense that adversarial perturbations are purposely designed to deceive the classifier whereas experimental noises are uncontrolled imperfections occurring naturally in an experiment. In general, the experimental noises will not deceive a classifier and cannot be confused with adversarial perturbations. What we find in this work is that, the experimental noises are more likely to “act as” adversarial perturbations when a larger percentage of the input data are dropped or unavailable for the neural network-based classifiers. We believe that this finding is important because it highlights a trade-off between “saving experimental data (thus cost)” and the classification accuracy, thus would provide a valuable guide for learning phases of matter with neural networks in practice.

In this revised manuscript, we have clarified this point and explicitly stated in the abstract and introduction that the experimental noises “act as” adversarial perturbations. We have also carefully revised several related statements to avoid possible confusions.

Comment 4 of Reviewer #1:Furthermore, though less critical, I would suggest that the authors carefully review the use of proper scientific English in their manuscript. I will not go into the details of the whole paper, which requires a proper review by the authors on the language used, but for example, the modification in the abstract states: “We show that the experimental noises will be more likely to become adversarial perturbations when the neural net-based classifiers present more advantages on data efficiency. We imple-

ment experimentally several adversarial examples which can deceive the splendid phase classifier with a notably high confidence, while keeping their physical properties unchanged.” Not only the language needs to be improved, but also the authors should avoid, in the whole manuscript, using extremely qualitative and colloquial terms such as “splendid”. Also, what are the physical properties left unchanged here, those of the system or those of the classifier? What do we mean in the context of the abstract for more advantages? It is somewhat a vague and unclear sentence. Other parts of the manuscript present similar problems.

Authors’ response: We thank the reviewer for going through the manuscript carefully and for raising concerns about the presentation of the manuscript. Based on his/her suggestions, we have carefully reviewed the whole manuscript and revised a number of statements so as to avoid possible confusions. We have removed all extremely qualitative and colloquial terms, such as “notably”, “splendid”, etc. Now, the presentation of the manuscript has been significantly improved.

In summary, we greatly appreciate the reviewer’s valuable suggestions/comments, which have guided us to further improve the manuscript substantially. We believe that this work is a timely and important contribution to the rapidly growing interdisciplinary field of machine learning phases of matter. We hope that the strengthened manuscript and our responses will satisfy the reviewer and our manuscript can be published in Nature Communications.

Response to Reviewer #2:

Comment of Reviewer #2:

I have read the revised manuscript and the reply written by the authors. The main strength of the paper is that the authors have experimentally demonstrated an adversarial attack for the classification of a topological index despite a very high fidelity. They also show that experimental noise could be relevant for such attacks in certain cases. Although the concept of adversarial attacks is certainly not new, its casting in an experimental context with a careful analysis is important. The presentation of the paper has much improved. The reply looks convincing to me. I recommend publication of this paper.

Authors’ response: We greatly appreciate the reviewer’s kind recommendation of our manuscript for publication in Nature Communications. We thank again for his/her valuable suggestions and comments in the first round, which helped us improve our manuscript significantly.

Response to Reviewer #3:

Comment 1 of Reviewer #3: I have carefully read the response of the authors to my report as well as to the other referee’s reports and the revised manuscript. The authors have extensively replied and reacted to my comments and questions. In particular, they have added a completely new analysis of the potential of experimental noises as adversarial examples, which I find very interesting and highly relevant. As more and more machine learning techniques are applied to experimental data, the influence of experimental errors/noise becomes a pressing question that needs to be addressed in order to be certain about what is being learned. The authors make an important step in this direction with their revised manuscript. I recommend

publication in Nature Communications.

Authors' response: We greatly appreciate the reviewer's very positive evaluation of our work and kind recommendation of our manuscript for publication in Nature Communications. We thank the reviewer for pointing out: "As more and more machine learning techniques are applied to experimental data, the influence of experimental errors/noise becomes a pressing question that needs to be addressed in order to be certain about what is being learned."

Comment 2 of Reviewer #3: Minor comments/typos:

Introduction, p.1: "In this work, we find that the adversarial examples are likely to occur by experimental noises instead of carefully crafted perturbations when the classifier presents more advantages on data efficiency." – I find this sentence somewhat confusing: I would think adversarial examples occur equally by experimental noises and carefully crafted perturbations? And 'presents more advantages on data efficiency' is somewhat vague, could be more specified.

p.1: "We then show that, even our legitimate samples have very high fidelity (99.7% on average)," – I think there might be a 'though' missing (even though our legitimate samples..)?

p.2: "We experimentally implemented the adversarial examples based on legitimate samples without data dropping." This sentence is not clear to me: what is happening now?

Fig.1c: why is the green line ($P(\chi^2=2)$) always zero?

Fig.1d: the y-axis label seems to appear twice.

p.5 "typical in experiments would nullify the adversarial examples. In the worst case" – space missing after full stop.

p.6 "As a possible solution to this problem, We tried a variation of the adversarial training" (We – > we)

p.6 "As a possible solution to this problem, We tried a variation of the adversarial training, which takes both carefully designed adversarial examples and randomly dropped data into account. While this defense strategy can indeed improve the robustness to carefully designed perturbations and experimental noises when a large portion of data are dropped, the classifier's performance near the phase transition points get much worse." – it's not entirely clear to me whether this is something that has been done/discussed in the main text before the Discussion section? If yes, I didn't realize it when reading the manuscript and the authors might want to consider to highlight it a bit more.

Authors' response:: We thank the reviewer for his/her very careful reading of our manuscript and for raising these comments/typos to our attention. Based on his/her report, we improved the presentation of the manuscript and clarified some important points. In the following, we address these comments/typos, one by one:

- We have improved the presentation of the Introduction. We have replaced the mentioned sentence with "In this work, we find that the experimental noises are more likely to act as adversarial perturbations when a larger percentage of the input data are dropped or unavailable for the neural network-based classifiers."
- We have added the word "though" and corrected this typo.
- We first numerically generate the theoretical adversarial examples based on legitimate samples, and then implement these generated adversarial examples (containing $10 \times 10 \times 10$ density matrices) in experiment. We have improved the presentation in the main text to avoid possible confusion.

- The green line ($P(\chi = -2)$) is not always zero. As the ratio of dropped data increases, the classification confidence for $\chi = -2$ increases to 0.0243 when 90% of the data are dropped.
- One y-axis label is from Fig. 1c and the other is from Fig. 1d. We have adjusted the Figure 1 to avoid this confusion.
- The corresponding typo has been corrected.
- The corresponding typo has been corrected.
- We thank the reviewer for this helpful suggestion. Indeed, we have studied a defense strategy, namely adversarial training, to increase the robustness of the phase classifier against adversarial perturbations. Due to space limitation, we have left the details to Supplementary Note 8. In this revised manuscript, we have added a new paragraph on adversarial training before the Discussion section to highlight related results.

In addition, we have carefully reviewed the whole manuscript and revised a number of other expressions that may cause confusions. Now, the presentation of the manuscript has been improved substantially.